# Automated data extraction from historical city directories: The rise and fall of mid-century gas stations in Providence, RI

Samuel Bell[1,2]*, Thomas Marlow[3], Kai Wombacher[1], Anina Hitt[4], Neev Parikh[4], Andras Zsom[1], Scott Frickel[3]

1 Advanced Research Computing, Center for Computation and Visualization, Brown University, Providence, Rhode Island, United States of America, 2 Planetary Science Institute, Tucson, Arizona, United States of America, 3 Institute at Brown for Environment and Society, Brown University, Providence, Rhode Island, United States of America, 4 Brown University, Providence, Rhode Island, United States of America

* sbell@psi.edu

**Data Availability Statement:** All relevant data are available for download from the Brown Digital Repository (https://doi.org/10.26300/typ4-nj27),

## Abstract

The location of defunct environmentally hazardous businesses like gas stations has many implications for modern American cities. To track down these locations, we present the *directoreadr* code (github.com/brown-ccv/directoreadr). Using scans of Polk city directories from Providence, RI, *directoreadr* extracts and parses business location data with a high degree of accuracy. The image processing pipeline ran without any human input for 94.4% of the pages we examined. For the remaining 5.6%, we processed them with some human input. Through hand-checking a sample of three years, we estimate that ~94.6% of historical gas stations are correctly identified and located, with historical street changes and non-standard address formats being the main drivers of errors. As an example use, we look at gas stations, finding that gas stations were most common early in the study period in 1936, beginning a sharp and steady decline around 1950. We are making the dataset produced by *directoreadr* publicly available. We hope it will be used to explore a range of important questions about socioeconomic patterns in Providence and cities like it during the transformations of the mid-1900s.

## 1. Background

Until the passage of the Resource Conservation and Recovery Act (RCRA) of 1976, waste produced during commercial and industrial activities in the United States was largely unregulated [1]. It took another decade until programs like the Environmental Protection Agency's (EPA) Toxic Release Inventory [2] were established for keeping track of emissions from the largest and most hazardous facilities. These regulatory dynamics, combined with businesses' tendencies to constantly churn in and out of operation over time, has created an urban environment covered with the relic sites and toxic legacies of past economic activity [3]. This is a serious concern for both community members worried about their health [4] and regulators and environmental professionals interested in locating and remediating contaminated sites.

and the *directoreadr* code are available at github.com/brown-ccv/directoreadr.

**Funding:** We acknowledge support from the Institute at Brown for Environment and Society, which funds a Research Assistantship for T.M., and from the Superfund Research Program of the NIEHS grant 2P42 ES013660. This work has also benefited from seed grants from the Brown University Office of the Vice President for Research (grant GR300065) to SF and the Brown Social Sciences Research Institute to SF and AZ. The funders did not play a role in the study design, data collection and analysis, decision to publish, or preparation of the manuscript.

**Competing interests:** The authors have declared that no competing interests exist.

Unfortunately, many relic sites of historical businesses with modern environmental implications remain untracked and unknown, with nearby residents unaware. As a result, there is a great need for accurate historical business location data dating back to before regulatory agencies began tracking these issues. To address this problem, we present *directoreadr*, a new code for extracting historical business data from scans of city directories.

Previous work focused on developing a software pipeline for processing historical directories specific to industrial manufacturing [5]. The result was the *georeg* code (https://github.com/brown-ccv/georeg), which was able to process digitized industrial directories to produce a near comprehensive dataset of industrial site locations and activities in Rhode Island for the years 1953–2012. This has been used productively for a range of scientific and community activities [6]. However, while industrial production is a major source of urban pollution, it represents only a selection of economic activities that leave behind on-site contaminants. Gas stations are another such commercial activity of concern. According to the EPA, underground gas and oil storage tanks at these sites are a leading source of groundwater contamination [7]. And while the federal government has been monitoring underground storage tanks (USTs) since the mid 1980s through RCRA, older USTs are added to lists only as they are discovered.

Therefore, in the current paper we develop an approach to collecting the historical location of commercial sites from city directories. Since the 1930s, the Polk Corporation has maintained detailed city directories for most American cities. Compiled annually, these books contain a comprehensive list of area businesses in the yellow pages.

Because the structure of the data in the city directories was considerably different from the industrial registries, we have developed a new code, *directoreadr*, instead of adapting the *georeg* code of [5]. We did use the same custom geocoder as [5], although the geocoding processing code was quite different. With the new algorithms in the *directoreadr* code, we are able to efficiently process data with a substantially higher degree of accuracy than [5].

To develop and test *directoreadr*, we have focused on city directories from Providence, Rhode Island. Using the scanned images of these directories, *directoreadr* is able to extract a company's name, address, and business type. These data are then geocoded to provide latitude and longitude. To show an example use of these data to examine environmentally hazardous sites, we focus on gas stations, but the applications are not limited to tracking environmentally hazardous sites. These data can answer many important research questions across a range of topics that are of interest to many social science and environmental disciplines, from economics to ecology.

## 2. Data

For the purposes of this project, we have focused on the yellow pages business directory within the city directories. We examined 27 city directories from the city of Providence, RI with dates from 1936 to 1990. Beginning in 1940, the city directories were produced by the Polk corporation, but the three city directories from before 1940 were produced by the Sampson and Murdock corporation. By extracting this detailed spatio-temporal business data, we allow for socioenvironmental analysis of changes in the land use of industrial sites, manufacturing zones, or other potentially hazardous areas, such as current and former gas station sites.

Digitization was performed by the Internet Archive's office at the Boston Public Library, and the physical books were supplied by both the Boston Public Library and the Providence Public Library. The Internet Archive uses a standardized digitization process, delivering 300 dpi 8-bit color images with a lossy compression in the wavelet-based JPEG 2000 format. We convert these files to grayscale and do not use color information. The raw scans are available

online from the Internet Archive, and we provide the links in the *directoreadr* GitHub ([github. com/brown-ccv/directoreadr](github.com/brown-ccv/directoreadr)).

## 3. Methods

As input, *directoreadr* takes a series of page images; as output, *directoreadr* produces a database of businesses and locations, along with error files containing dropped addresses, geocoder errors, and addresses in another city. The pipeline consists of a series of discrete processing steps: grayscale thresholding, ad removal, margin cropping, column chopping, line chopping, Optical Character Recognition (OCR), header identification, entry concatenation, text cleaning, address parsing, street matching, and geocoding.

### 3.1 Image preprocessing

**3.1.1 Grayscale thresholding.**   The original color images are read into *directoreadr* as 8-bit grayscale images with an integer pixel value ranging from 0 to 255, and the first step of the pipeline is to convert these images to a binary format, where each pixel value is either 0 or 1. This binarization step enables us to detect connected areas of black pixels, a core component of many of the computer vision algorithms we use. To do this, we use a fixed threshold across the entire page, which *directoreadr* estimates from the page's pixel value distribution. Incorrect grayscale thresholds are one of the largest sources of error in *directoreadr*, and better grayscale thresholding would provide more accurate results.

**3.1.2 Ad removal.**   After producing the binary images, we remove the advertisements along the border of the pages, as well as lines and decorations within and between the columns of text. For the sake of simplicity, we refer to all page features to be removed as "ads." To identify and separate out the ads from the text, we leverage two different geometric characteristics: First, the ads tend to be outlined by simple shapes like direct rectangles. Second, the ads tend to be much larger in extent than the characters of the text. Fig 1 shows an example of ad removal.

Using the OpenCV *contours* method, we identify regions of connected pixels. For each pixel contour, we calculate both the perimeter of the contour and the perimeter of the bounding box, the smallest possible horizontal rectangle circumscribing the contour. In most cases,

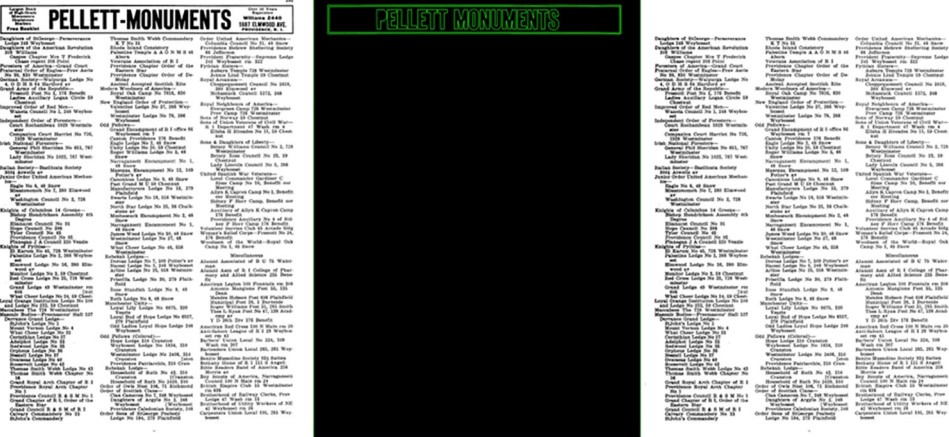

**Fig 1. An example city directory page showing the ad removal process.** The first panel shows the city directory page as a binary image. The second shows the contours identified as ads by the ad removal algorithm. The final panel shows the image after ad removal.

ads can be separated from text simply by looking at the perimeter of the bounding box. However, in a few cases, many characters of text blur into each other, and the perimeter of the bounding box can be as large as it is for the smallest ads. To address this, we multiply the perimeter of the bounding box by the ratio between the perimeter of the bounding box and the perimeter:

$$p_{adjusted} = \frac{p_{bounding}^2}{p}.$$

Because text has a more complex shape than the ads, the ratio of bounded perimeter to contour perimeter is much lower for text than for ads, and it helps separate the text from the ads. Once we have identified the contours around the ads, we remove any black pixels within the bounding box around those contours.

In most cases, the ads around the edge of the page are surrounded by horizontal rectangles. When they aren't, the next step is to identify where the columns of ads are and remove all black pixels there. Even then, in a few cases, ad removal fails, and the image has to be cropped by hand.

**3.1.3 Margin cropping.** Once the ads have been removed and replaced by whitespace, the columns of text in the center of the page are still surrounded by whitespace. To focus in on the text, the next step in the *directoreadr* pipeline is to remove the whitespace. To allow for specks, lines, and other noise on the page, we set a pixel threshold. Margin cropping is fairly straightforward and rarely creates problems.

## 3.2 Image segmentation

**3.2.1 Column chopping.** Each page is set up with columns of text (usually three columns), and in order to preserve information about text location, we separate the text into the columns. To identify the column breaks, we sum up the number of black pixels in each vertical line of pixels in the image. Around the column breaks, there are dips in the number of black pixels. To identify the location of the dips, we set a pixel threshold and identify the vertical lines with fewer black pixels than the threshold value. Using the mean-shift algorithm, we cluster those vertical lines. Unlike traditional clustering algorithms, like k-means, which take a number of clusters as an input, mean-shift figures out the optimal number of clusters. As the cut point for column separation, we pick the right-most vertical line in each cluster.

One of the key features of this algorithm is to err on the side of failure, throwing an error when the ad removal has performed poorly. The goal of this design is to allow for hand-chopping when it will meaningfully improve the results, and for all of the failure cases, we generated the columns through hand-chopping.

**3.2.2 Line chopping.** Once we have the columns of text, we then chop the columns into individual lines of text. To identify the lines of text, we use a similar process to identifying the columns. We calculate the number of pixels in each horizontal line of pixels in the column. Then, we cluster the horizontal lines of pixels that fall below the pixel threshold, using mean-shift to identify the entry breaks. If there are large blocks of entries that don't separate, we then run the algorithm on them with a higher black pixel threshold. This higher threshold is typically necessary when the page image is warped or tilted. Highly robust, this process rarely produces errors unless there are more serious problems with the image.

## 3.3 Address processing

**3.3.1 OCR.** Entering this part of the pipeline, we have a directory of images where each image represents a single line from one of the columns on the page. To convert these images of

text to a string of text, we use the Tesseract OCR package developed by the Google corporation [8]. OCR is not perfect, and it does produce some errors, so downstream text parsing parts of *directoreadr* must account for these errors.

**3.3.2 Header determination.** The data in the city directories are grouped under headers that describe the type of business, and these headers must be identified. Depending on the year, the city directories identify headers using a number of different characteristics. Headers are typically indented, and they sometimes contain all caps. Often, headers have asterisks before them. Depending on the year, *directoreadr* selects from five different header determination algorithms, most of which center around how many pixels each line is indented by. Because some columns are tilted, we calculate a relative indentation compared to nearby lines. Our header detection algorithms relied on indentation and capitalization as the primary detection features, and we did not build a robust header algorithm for 1964, the one year in which headers were not indented, and both the headers and the text were in all caps. As a result, most header identifications for 1964 are incorrect.

**3.3.3 Entry concatenation.** In many cases, entries in the columns of text are too long for one line and continue onto the next line. In all of these cases, the next line is indented, but not by as much as a header is. Using the indentation data, we concatenate the multi-line entries into single strings of text.

**3.3.4 Text cleaning.** Most of the raw entries just contain a business name and an address, but some of them contain additional information that must be removed, like a telephone number or a floor or room number. To clean these data, we used a complex series of pattern matching operations. In some cases, especially in older books, there were multiple addresses for a business in a single entry, and we split these lists based on the positions of commas and the word "and."

**3.3.5 Address parsing.** We start by using regular expressions to search the entry's string of text for abbreviations like: St., Av., Ct., Dr., Rd., Ave., and Ln. in either upper or lower case. If one of those abbreviations is detected in the string, the algorithm searches for a group of digits before the abbreviation. First, it classifies the string of text between the number and abbreviation as the address, and then it classifies the text before the address number as the company name. If the abbreviation is not detected, the algorithm will still try to parse out the address by searching for the address number and classifying the string of text after the number as the address, then classifying the string proceeding the number as the company name.

This parsing algorithm is not perfect. For instance, it requires digits, not spelled out numbers, for the address, and it does not work for addresses written as the corner between two streets instead of a numbered address. However, it is generalized enough to work well across many different formats because it is built on simple, consistent address components.

**3.3.6 Street matching.** Because a number of the streets contained OCR errors, we used fuzzy matching to produce true street names. We developed two lists of streets, a list of current streets and a list of historical streets. The historical street list was developed through hand examination of historical maps and is not fully comprehensive. Because we only had a database of Providence streets, we removed the addresses we could identify as belonging to another Rhode Island municipality.

Using the *fuzzywuzzy* package in Python, we created a scoring algorithm to quantify how close an OCR reading of a street name is to a street in the true street name list. This scoring algorithm is based off the Levenshtein distance ratio:

$$ratio(s1, s2) = 100 * \left(1 - \frac{D(s1, s2)}{L(s1) + L(s2)}\right),$$

where *s1* and *s2* are the two strings being compared, *L* is a function giving the length of a string and *D* is a function giving the Levenshtein distance between two strings. The Levenshtein distance is the minimum number of edit operations (substitutions, deletions, or additions) required to convert one string into another. For instance, the Levenshtein distance between "park" and "barks" is 2, one substitution and one addition. The ratio would be $100^*(1\text{-}2/9) = 77.8\%$. The scoring algorithm is:

$$score = \frac{ratio(s1, s2) + ratio(LongestWord(s1), LongestWord(s2))}{2},$$

where *LongestWord* is a function that gives the longest word of a string. The reason for adding additional emphasis on the longest word was to emphasize the core street name. For instance, we wanted "BROADWAY" to match with "BROADWAY ST."

For each street in the true streets list, we calculated the matching score between the OCR result and the street in the true streets list, selecting the true street with the highest score. To guard against false positives, we removed any matches with a score below 80. One of the *directoreadr* output files lists any entries that failed to match to a known street, failed to parse as an address, or contained empty text. If desired, these files could be examined for hand coding, but we do not do that here.

To dramatically improve the speed of the street search code, we only searched for unique queries, saving each input string in a dictionary to convert it to its street match. That way, once we had done the street searching for one OCR reading of a street, when that same OCR reading came up again, we wouldn't have to repeat the search.

**3.3.7 Geocoding.**   The last component of the *directoreadr* pipeline is geocoding the cleaned and parsed addresses to obtain the latitude and longitude coordinates of the businesses. After researching several different geocoding options, from paid services (SmartyStreets) to free APIs (Google Maps), we decided to implement the geocoder built for [5] using ArcGIS software, with data from Rhode Island's E911 database. This geocoder was free for our use, given that Brown University had an in-house ArcGIS server, but because geocoders are proprietary, in our published version of the code, we do not include the api key necessary to run the geocoder.

To improve the speed of the geocoding, we ran 50 concurrent searches and searched only for unique addresses, building up a dictionary of geocoder results to reference in future runs of the program. Because many addresses were repeated across many years, this drastically sped up the process.

The geocoder only contained data on current street layouts. Providence, however, like many American cities, has seen considerable change in its street pattern over the course of the study period. Many streets have been wholly or partially demolished, and others have been renumbered. To address this problem and prevent against false positives, we utilized the geocoder confidence score, and we removed any addresses with a confidence score under a perfect score of 100 from our final results. This way, we only accepted matches that were a perfect match to a known current address in our geocoder's address database. To address the most common of these addresses, we allowed for hard-coding of hand-identified historical geocodes, entering hard-coded locations for four large buildings with many businesses at those addresses.

## 4. Results and discussion

The image processing portion of the pipeline had a success rate of 94.4%. In our dataset, we ran the algorithm on 2,582 individual pages. For these pages, 144 or 5.6% required hand-

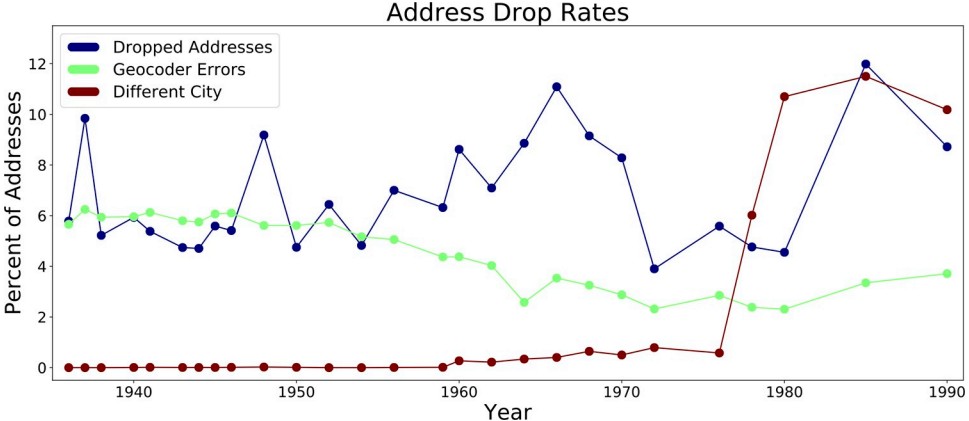

**Fig 2. Address drop and geocoder error rates by year.**

chopping in order to process. We designed the column-chopping algorithm to deliberately fail when there were likely errors with the ad removal algorithm. The goal was to require hand-chopping whenever it would meaningfully improve the end result, not just in cases where it was strictly necessary. Because of the hand-chopping, we were able to pass all of the pages through to the OCR and text parsing algorithms.

In the text parsing algorithm, 6.7% of all entries were dropped as not a successfully identified and matched address. These include both entries that should be dropped and entries that were dropped because of an error. In 38.2% of these cases, the algorithm failed to parse an address at all. In 10.3% of these cases, the algorithm parsed an address but returned an empty string for the street. In 4.2% of these cases, the algorithm parsed an address but threw an error in street matching. And in 47.2% of these cases, the algorithm successfully parsed an address and matched a street, but the confidence score was too low for us to be sure the address was correct. In some of the address drop cases, the addresses were outside of Providence, sometimes outside of Rhode Island. Others reflected an idiosyncratic address form the algorithm wasn't set up to parse. For instance, some addresses were named buildings without an address (e.g. "Arcade Bldg" or "Industrial Trust Bldg"). Others were street corners instead of numerical addresses. Of course, many of the address drop cases represented failures of the OCR or failures of the header identification, concatenation, and entry chopping algorithms. Address drop rates were not strongly correlated with time (Fig 2).

The geocoder algorithm produced errors in 4.7% of cases. (Errors here are defined by failing to produce a prefect geocoder confidence score.) Unsurprisingly, these errors were higher in earlier years when the Providence street pattern was considerably different (Fig 2). Towards the end of the study period, the percentage of addresses outside of Providence increased sharply. While we were able to capture most of these, we were not successful in all cases. Many addresses from a different city were not recognized as belonging to a different city, and when they were processed, they led to dropped addresses or geocoder errors.

These statistics only capture the places where the code generated errors or flags. In order to fully assess the ultimate accuracy of the code, we hand-examined the error rate for gas stations in three years: 1936, 1962, and 1990. In 1936, 220 out of 242 gas stations were correctly identified, for an accuracy rate of 90.9%. Of the 22 missing gas stations, eleven were missing due to geocoder errors cause by historical street changes, seven were missing because of non-standard address formats that *directoreadr* could not parse correctly, one had the wrong address read, and only three were entirely missing. In 1962, 219 out of 224 gas stations were correctly

identified, for an accuracy rate of 97.8%. Of the five missing gas stations, one was dropped, and four had a geocoder error. Three out of the four geocoder errors represented cases where the addresses had been read incorrectly. In 1990, 71 out of 73 gas stations were correctly identified, for an accuracy rate or 97.3%. Both of the missing gas stations were dropped. We define "correctly identified" as cases where the algorithm correctly read and parsed the address, identified the right street, and geocoded it correctly. These statistics do not include errors in correctly reading and parsing the business names.

The total accuracy rate from the gas stations in the three years we examined by hand was 94.6%. (This number is somewhat skewed because 1936 had the most gas stations. The average of the three accuracy rates was 95.3%.)

Of all the errors we identified during our hand validation, only one was a false positive, a case where an entry was processed without errors but returned the wrong address and geocoded location. This false positive occurred in as part of a long list of locations of the Socony-Vacuum Oil Company where two entries were merged together. The section of the text read "269 Valley, Waterman cor E River," which got parsed as one address at 269 Waterman St., which geocoded successfully. In this case, we recorded this as one station missed because of a non-standard address format and one address incorrectly parsed as a different address.

We are making the data available for download from the Brown Digital Repository (https://doi.org/10.26300/typ4-nj27), and we are making the *directoreadr* code available at github.com/brown-ccv/directoreadr.

## 4.1 False positives

We did our best to design our algorithm to avoid false positives at all cost, leading to the errors being overwhelmingly false negatives. However, it is probable that there are some other types of rare false positives we could not identify in our hand validation sample. For instance, if a street was renumbered, and an old address moved its location, the geocoder only knows the relationships between addresses and locations in the present day, and it would identify the old address with its present location. Without extensive historical research of every address, it is impossible to rule out that this may have happened with a very small percentage of our addresses. Additionally, a small portion of addresses outside Providence were not successfully identified as belonging to another city. While the vast majority of these addresses will not match to an exact address in Providence, a small percentage, mostly on common street names, will have an address in a different city that exactly matches a real address in Providence. No examples of these sorts of false positives were observed in our hand validation, but it is certainly probable that there may be a few examples of these sorts of false positives outside our hand validation sample.

## 4.2 Efficiency

Once the geocoding, street matching, and OCR results have been cached, the parsing algorithm runs in roughly 20s per book on a standard laptop, enabling faster debugging and development. With no cached results, the full *directoreadr* pipeline still runs in under 30 minutes per book. Before our efficiency improvements, *directoreadr* would take many hours to process a single book.

## 4.3 Example hazardous site: Gas stations

Because of their environmental importance, we selected gas stations as an example hazardous site type. Gas stations in Providence typically developed along main roads, avoiding wealthier neighborhoods like Providence's East Side (Fig 3).

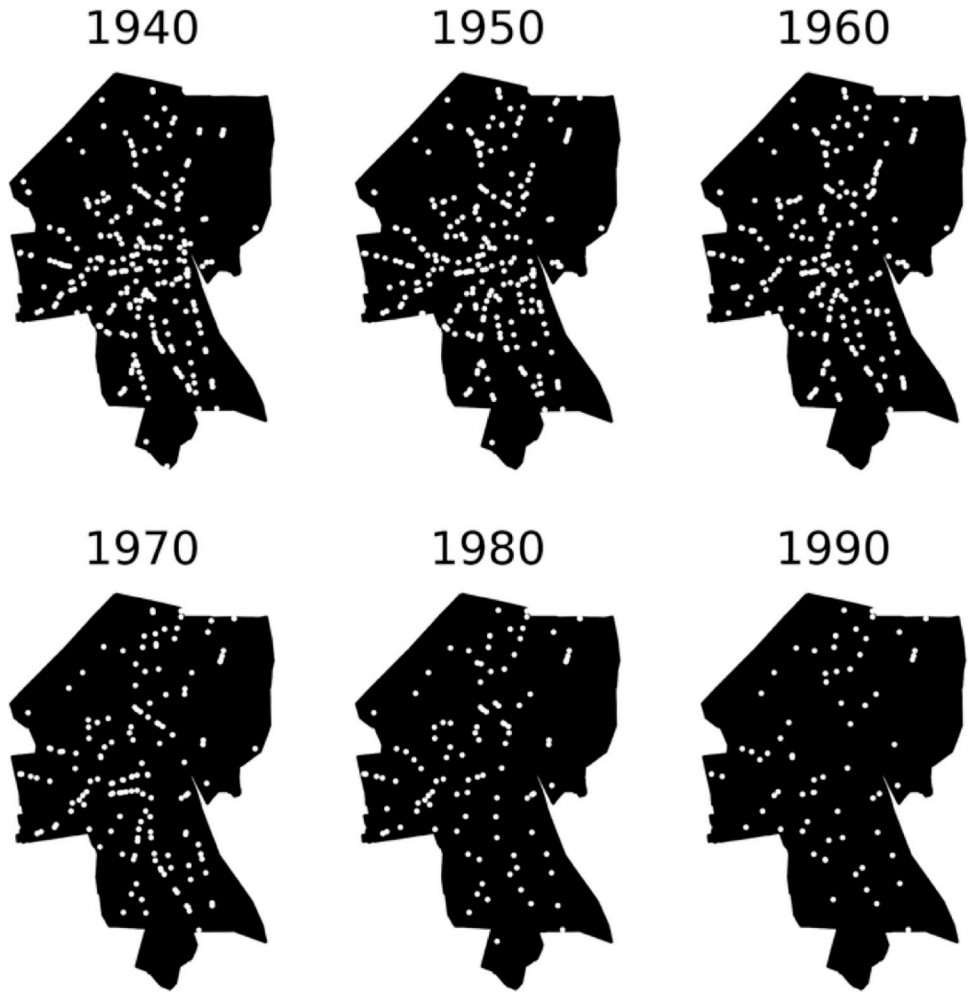

**Fig 3. Gas stations mapped in Providence from 1940 to 1990.**

Starting in the 1950s, gas stations began a precipitous decline in Providence (Figs 3 and 4). By 1990, there were only 75 gas stations in the city, a decline of 71% since 1950, when city

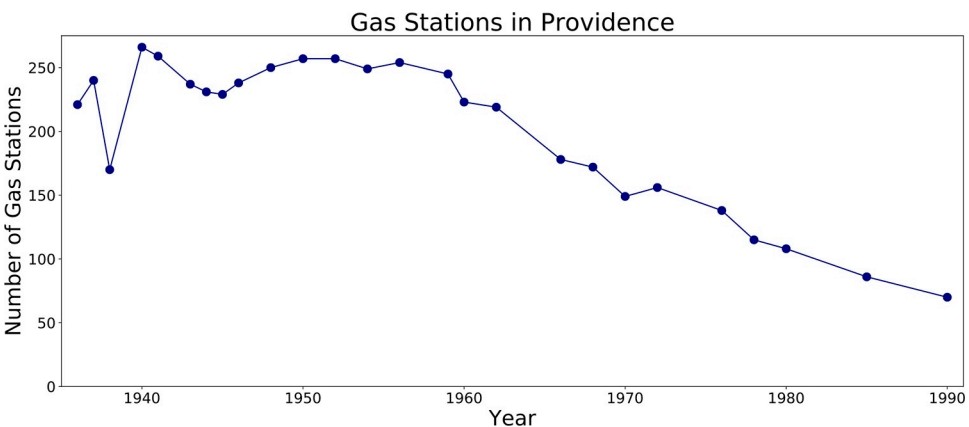

**Fig 4. Total number of gas stations recorded in Providence by *directoreadr*.**

directories list 257. This drop corresponds with a decline in the city's population, which dropped by a third between 1950 and 1980, the combined result of job loss from deindustrialization and displacement of minority residents whose neighborhoods were cleared for several ambitious "urban renewal" projects [9–13]. These changes were part of broader national trends of suburbanization and economic decline in the urban core [14–16]. Other factors specific to the service station and automobile industries also may have played a role. Broader regulatory changes likely also affected gas station counts, with zoning having a particularly important effect [17–19]. Because the rate of geocoder errors was higher in the earlier years, these figures probably underestimate the dramatic drop in the number of gas stations. Overall, we identified 526 unique gas station addresses in Providence over the study period, compared to just 114 gas station addresses recorded in the Rhode Island Department of Environmental Management Underground Storage Tank (UST) database (http://www.dem.ri.gov/programs/wastemanagement/inventories.php).

## 4.4 Applicability to other cities

Although we have developed the *directoreadr* code on directories from Providence, RI, these directory formats are fairly similar in different cities, and *directoreadr* should be easily adaptable to cities all across the country. The Polk Corporation produced comprehensive sets of city directories for nearly all major and mid-sized American cities, and the formats are largely similar from city to city. While some modifications to the code may be necessary to address small differences in layout, heading design, text abbreviations, and ad shape, the modifications needed should be small. While archival and permissions work will always be necessary to assemble and scan the books, these city directories have been preserved and maintained by libraries around the country, and a comprehensive set of city directories should be available at a feasible cost.

## 5. Conclusions

We have successfully built the *directoreadr* code: a pipeline for the digitization, extraction, and processing of city directory data. We were able to process 94.4% of the pages without any human input, and the rest required only minor human input. The code correctly processed, identified, and located 94.6% of gas stations in our hand validation set. We have developed and tested the code on city directories from Providence, RI, and it should be adaptable to cities all across the country with minimal modifications. There are many potential uses of these data, and we have demonstrated mapping of environmentally hazardous historical gas station sites as an example. We found that gas stations were most plentiful in the 1930s and 1940s, with a dramatic decline beginning around 1950. We identified 526 unique historical gas station sites in Providence, compared with just 114 in the Rhode Island Department of Environmental Management's Underground Storage Tank database.

## Acknowledgments

We would like to thank the Providence Public Library and the Boston Public Library for providing many of the physical directories for scanning, as well as the Internet Archive for doing the scanning. We would like to specially thank Kate Wells of the Providence Public Library for her invaluable advice and assistance in arranging the scanning.

## Author Contributions

**Conceptualization:** Thomas Marlow, Kai Wombacher, Andras Zsom, Scott Frickel.

**Data curation:** Samuel Bell, Kai Wombacher.

**Formal analysis:** Samuel Bell, Thomas Marlow.

**Funding acquisition:** Andras Zsom, Scott Frickel.

**Investigation:** Samuel Bell, Thomas Marlow, Scott Frickel.

**Methodology:** Samuel Bell, Thomas Marlow, Kai Wombacher, Andras Zsom.

**Project administration:** Scott Frickel.

**Resources:** Scott Frickel.

**Software:** Samuel Bell, Kai Wombacher, Anina Hitt, Neev Parikh, Andras Zsom.

**Supervision:** Andras Zsom, Scott Frickel.

**Validation:** Thomas Marlow.

**Visualization:** Samuel Bell, Thomas Marlow.

**Writing – original draft:** Samuel Bell.

**Writing – review & editing:** Samuel Bell, Thomas Marlow, Anina Hitt, Neev Parikh, Andras Zsom, Scott Frickel.

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
