## [Decision Letter · Decision Letter 0]

9 Sep 2019

PONE-D-19-19315

Automated data extraction from historical city directories the rise and fall of mid-century gas stations in Providence, RI

PLOS ONE

Dear Dr. Bell,

Thank you for submitting your manuscript to PLOS ONE. After careful consideration, we feel that it has merit but does not fully meet PLOS ONE’s publication criteria as it currently stands. Therefore, we invite you to submit a revised version of the manuscript that addresses the points raised during the review process.

I have also read your paper and I like it. Permit me to share a few thoughts.

The paper addresses an important roadblock in research questions investigating historical microdata, the time-consuming task of transcribing large amounts of data. In the past, this arduous task has been delegated to research assistants, interns, and more recently crowdsourced through “digital volunteers” or “micro-volunteers”. The paper also highlights the issue of the toxic legacy of buildings prior to 1976, which may be unknown as land use patterns change over time.

The methods laid out in the paper are sound and should address the majority of steps needed to process historical directories. However, a main question is: how viable is the method for smaller numbers of records? For under 2,500 records, a research assistant coder could probably complete the same task in a few days. There are a variety of ways that errors could enter the processing pipeline, leading to necessary double checking that would be duplicative of the work of coders. The larger the dataset and more years investigated, the more this method becomes viable. Image quality seems to be a major factor in the utility of this method and good quality scans of well-maintained historical documents are crucial. The development of this method is promising and demonstrates the future possibilities and efficacy of machine learning in converting historical paper databases to useful microdata for researchers.

In your revision, please try to incorporate the reviewer's points and my discussion. I look forward to reading your revision.

We would appreciate receiving your revised manuscript by Oct 24 2019 11:59PM. To enhance the reproducibility of your results, we recommend that if applicable you deposit your laboratory protocols in protocols.io, where a protocol can be assigned its own identifier (DOI) such that it can be cited independently in the future. For instructions see: http://journals.plos.org/plosone/s/submission-guidelines#loc-laboratory-protocols

We look forward to receiving your revised manuscript.

Kind regards,

Matthew Kahn

Academic Editor

PLOS ONE

Journal Requirements:

1. We note that you have stated that you will provide repository information for your data at acceptance. Should your manuscript be accepted for publication, we will hold it until you provide the relevant accession numbers or DOIs necessary to access your data. If you wish to make changes to your Data Availability statement, please describe these changes in your cover letter and we will update your Data Availability statement to reflect the information you provide.

Reviewers' comments:

Reviewer's Responses to Questions

**Comments to the Author**

1. Is the manuscript technically sound, and do the data support the conclusions?

Reviewer #1: Partly

2. Has the statistical analysis been performed appropriately and rigorously? 

Reviewer #1: N/A

3. Have the authors made all data underlying the findings in their manuscript fully available?

Reviewer #1: Yes

4. Is the manuscript presented in an intelligible fashion and written in standard English?

Reviewer #1: Yes

5. Review Comments to the Author

Reviewer #1: I enjoyed this article and think that the process you’re describing has broad utility for social science researchers, and any other researchers that might want to rely on old text as data.

A. Validation

A key statistic in this paper is that about 95 percent of gas stations (94.6) are “correctly identified.” I was never entirely sure what this term meant, and that the baseline for “correct” was. On page 16, I think we learn that “success” does not mean that your method returns a result that is as good as hand-coding – which I think is the most natural definition of success. As far as I understand, “success” means that the processing runs without errors.

More generally, I think the paper would be tremendously stronger if you more generally considered

- true positives

- false positives

in addition to what you do already quite well, which is consider

- true negatives

- false negatives

Suppose you geocode an address “successfully,” but that location is wrong. This is a false positive. Do you have any sense of the scope of false positives in your data? You do a good job looking at negatives – I think the paper would be strengthened by looking at positives as well.

At multiple places in section 3, you write some version of “things fail and require correction by hand.” It would be very helpful to know

- how many times you had to do this

- what “fails” means. Does this mean there was an error and the image wouldn’t process? Or that you looked at the output and decided it was of too low quality? If the latter, what was your metric?

What do you do with the addresses that fail to match? Do you just leave them as unmatched? Or do you follow-up with hand coding?

On page 17, you write that “the geocoder algorithm produced errors in 4.7% of cases.” What is an error here? Where the program fails to code at all? Is a location outside Providence an error? And what about incorrect coding of addresses that are coded?

It would also be helpful to show the rate of failure by year (not the number, but the rate). This would help readers assess whether this technique is reasonable for time series data.

B. And what do you do with this?

At the end of the article, you give some tantalizing short sentences on how the number of gas stations has declined tremendously over time in Providence. It would be extremely interesting to understand the determinants of gas station depth.

C. Visualizations

You present a number of over-time charts. When showing data over time, it is both standard and helpful to use line charts. In your case, I’d use a line with dots; the dots will indicate where you actually have data, and the lines will allow us to follow the key patterns over time.

D. Writing

I found the overall organization and communication in this paper to be quite sound.

The writing more broadly would be substantially improved with the help of a professional editor. These edits would go a long way toward improving the readability of the paper and get your points across more effectively.

6. PLOS authors have the option to publish the peer review history of their article (what does this mean?). If published, this will include your full peer review and any attached files.

Reviewer #1: No

---

## [Author Response · Author response to Decision Letter 0]

23 Oct 2019

We have included more detailed responses to all reviewer questions in our response to reviewers file.

---

## [Decision Letter · Decision Letter 1]

30 Jan 2020

PONE-D-19-19315R1

Automated data extraction from historical city directories the rise and fall of mid-century gas stations in Providence, RI

PLOS ONE

Dear Dr. Bell,

Thank you for submitting your manuscript to PLOS ONE. After careful consideration, we feel that it has merit but does not fully meet PLOS ONE’s publication criteria as it currently stands. Therefore, we invite you to submit a revised version of the manuscript that addresses the points raised during the review process.

We would appreciate receiving your revised manuscript by Mar 15 2020 11:59PM. To enhance the reproducibility of your results, we recommend that if applicable you deposit your laboratory protocols in protocols.io, where a protocol can be assigned its own identifier (DOI) such that it can be cited independently in the future. For instructions see: http://journals.plos.org/plosone/s/submission-guidelines#loc-laboratory-protocols

We look forward to receiving your revised manuscript.

Kind regards,

Chuanwang Sun

Academic Editor

PLOS ONE

Reviewers' comments:

Reviewer #1: Thanks for the revisions; I liked the explanation about false positives.

On page 18, you write "three out if the four" which should be "three out OF the four".

I also feel that your remaining bar chart should be a line chart! But I'm not going to hold up publication over this.

Overall, I really enjoyed the your explanation about the process. It was clear and inspiring.

Reviewer #2: This paper applies Directoreadr code to track the past location of Providence and discusses the social and economic issues about Providence’s transition in the mid-90s. The topic of this paper is interesting and research-meaningful.

Generally speaking, the logic and method in the research are feasible. However, some questions should be paid attention to as an academic paper.

Firstly, the importance and innovation of this research should be presented in the introduction.

Secondly, the conclusion is too simple to demonstrate the details in the result. Meanwhile the reason why the research method is appropriate for the other cities isn’t stated in detail either.

---

## [Author Response · Author response to Decision Letter 1]

1 Jul 2020

Reviewers' comments:

Reviewer #1: Thanks for the revisions; I liked the explanation about false positives.

Great! Thanks so much!

On page 18, you write "three out if the four" which should be "three out OF the four".

Good catch! We’ve fixed this.

I also feel that your remaining bar chart should be a line chart! But I'm not going to hold up publication over this.

On balance, we feel that it is better to have a bar chart because it more clearly visually emphasizes our data coverage. We do not feel strongly about it, though, so we have happily switched to a line chart here.

Overall, I really enjoyed the your explanation about the process. It was clear and inspiring.

Thank you so much!

Reviewer #2: This paper applies Directoreadr code to track the past location of Providence and discusses the social and economic issues about Providence’s transition in the mid-90s. The topic of this paper is interesting and research-meaningful.

Generally speaking, the logic and method in the research are feasible.

Thank you!

However, some questions should be paid attention to as an academic paper.

Firstly, the importance and innovation of this research should be presented in the introduction.

We have added several new sentences to more definitively highlight the importance and innovation of the paper in the introduction.

Secondly, the conclusion is too simple to demonstrate the details in the result. 

We have written a longer and more descriptive conclusion. A short conclusion is a rhetorical decision we are not strongly committed to.

Meanwhile the reason why the research method is appropriate for the other cities isn’t stated in detail either.

We have expanded our discussion of this issue into a separate section.

---

## [Editor Report · Decision Letter 2]

6 Jul 2020

Automated data extraction from historical city directories the rise and fall of mid-century gas stations in Providence, RI

PONE-D-19-19315R2

Dear Dr. Bell,

We’re pleased to inform you that your manuscript has been judged scientifically suitable for publication and will be formally accepted for publication once it meets all outstanding technical requirements.

Kind regards,

Chuanwang Sun

Academic Editor

PLOS ONE